# Entanglement Witness for the Weak Equivalence Principle

**DOI:** 10.3390/e25030448

**Published:** 2023-03-03

**Authors:** Sougato Bose, Anupam Mazumdar, Martine Schut, Marko Toroš

**Affiliations:** 1Department of Physics and Astronomy, University College London, Gower Street, London WC1E 6BT, UK; 2Van Swinderen Institute, University of Groningen, 9747 AG Groningen, The Netherlands; 3School of Physics and Astronomy, University of Glasgow, Glasgow G12 8QQ, UK

**Keywords:** equivalence principle, entanglement, quantum gravity

## Abstract

The Einstein equivalence principle is based on the equality of gravitational and inertial mass, which has led to the universality of a free-fall concept. The principle has been extremely well tested so far and has been tested with a great precision. However, all these tests and the corresponding arguments are based on a classical setup where the notion of position and velocity of the mass is associated with a classical value as opposed to the quantum entities.Here, we provide a simple quantum protocol based on creating large spatial superposition states in a laboratory to test the *quantum regime of the equivalence principle* where both matter and gravity are treated at par as a quantum entity. The two gravitational masses of the two spatial superpositions source the gravitational potential for each other. We argue that such a quantum protocol is unique with regard to testing especially the generalisation of the weak equivalence principle by constraining the equality of gravitational and inertial mass via witnessing quantum entanglement.

## 1. Introduction

Over the last century, the general theory of relativity has passed a number of stringent experimental tests [1]. Among its core tenets, still experimentally unchallenged, is the Einstein equivalence principle (EEP). The EEP, in its modern form [2], consists of three parts: the universality of free fall, also known as the weak equivalence principle (WEP), local Lorentz invariance (LLI) and local position invariance (LPI). The WEP implies that all objects fall at the same rate, regardless of their internal composition or structure, as long as tidal effects can be neglected.

Generally, the equivalence principle states as follows: The equations of motion for matter coupled to gravity are locally identical to the equations of motion for matter in the absence of gravity.

In Newton’s theory of gravity (which is non-relativistic), WEP is phrased as the equality of the inertial and gravitational mass—the mass appearing in Newton’s second law and the Newtonian gravitational potential, respectively. In addition, LLI and LPI assume that any local non-gravitational experiment will give the same result, regardless of the velocity of the freely-falling reference frame in which it is performed and regardless of where and when the experiments are performed.

The WEP has been put under experimental scrutiny from the days of Galileo, and space-based experiments such as MICROSCOPE have placed stringent bounds on the universality of free-fall, constraining the value of the Eötvös parameter to one part in $10^{15}$ [3]. Experiments with trapped atoms and ions have tested possible LLI violations, parameterized by $\delta =|c^{-2}-1|$, with *c* the speed of light, confining the values of δ below $3\times 10^{-22}$ [4,5,6]. Atomic clock experiments, which test deviations from the gravitational red-shift formula, $z=\left(1+\alpha \right)\frac{\Delta U}{c^{2}}$, where *z* is the red-shift and ΔU the Newtonian potential difference between two clocks, have found that the LPI violation parameter α must be smaller than one part in $10^{6}$ [7].

The EEP is however formulated in an ostensibly classical framework, and its generalization to quantum mechanics requires careful considerations [8]. Quantum systems are described in terms of wave-functions, which do not have a point-like support and thus do not conform to the notion of test particles, central in the formulation of EEP. As first demonstrated by the Collela–Overhauser–Werner (COW) experiment [9], quantum experiments at the interface with gravity can no longer be described solely in terms of classical trajectories but require the computation of the quantum phases [10,11,12].

In the quantum domain, where the notion of particles and trajectories becomes vague, matter-waves act as *quantum probes* of the background gravitational field, requiring the generalization of the equivalence principle to the quantum domain [13,14,15,16,17,18,19], with ongoing experimental effort [20,21,22,23].

However, when the gravitational field is sourced by a quantum object, the assumption of a classical background of a gravitational field becomes problematic, and the EEP formulations discussed above cannot be directly applied. Some have even questioned the validity of EEP when *quantum sources* of gravity are involved [24,25], while some have pointed out its consistency and introduced generalized notions [26,27,28,29,30,31,32].

To date, there is no experimental evidence about the quantum-gravity interface in this regime, and any experiment shedding light on the gravitational field generated by a quantum source would be a major milestone [33,34,35].

The aims of this paper are to (i) introduce a generalized WEP capable of testing the equivalence between the inertial and the gravitational mass with quantum sources and quantum-natured gravity and (ii) provide a protocol for an experimental realization with matter-waves.

In this paper, we put forward a quantum protocol to test the equivalence between the inertial and the gravitational mass, where the gravitational mass is sourced solely by the two quantum systems. We work in the framework of perturbative canonical quantum gravity coupled to non-relativistic matter where such a problem can be unambiguously formulated and offer a pathway for experimental implementation with nano-particles. Using the ostensibly quantum notion of entanglement, we introduce the notion of the *entanglement entropy weak equivalence principle* (EEWEP), which can be tested by adapting the recently proposed quantum gravity induced entanglement of masses (QGEM) protocol [33]; see also [36] for a similar proposal. Unlike previous WEP tests, which relied on classical notions or single-particle interference, the EEWEP relies on two-particle entanglement—a hitherto unexplored regime of the quantum–gravity interface.

## 2. QGEM Scheme

Quantum mechanical sources of gravity pose significant conceptual questions and have led to several approaches to quantum gravity [37]. Here, we work within the framework of the perturbative low energy effective field theory of quantum gravity coupled to the non-relativistic matter-waves [38,39,40]. Within this framework, the QGEM protocol aims to test the *quantum nature for gravity* in a laboratory with the basic blueprint shown in Figure 1 [33,36]. In a nutshell, the two particles are placed sufficiently far apart that the electromagnetic interactions are negligible and at the same time close enough that the two particles become entangled through the gravitational interaction. The underlying mechanism for the generation of entanglement has been analysed within low energy effective field theory [34,35] and the framework of the Arnowitt–Desse–Meissner (ADM) approach [41], as well as in the path integral approach [42]. In the language of effective field theory of quantum gravity, at low energies, the two quantum systems are entangled due to the exchange of a graviton containing both the spin-2 and the spin-0 components, which are the dynamical off-shell degrees of freedom of a massless graviton in four dimensions [34,35].

The QGEM protocol, assuming the locality of interactions, results in an entanglement witness for the quantum character of gravity. The result is concurrent to the Local Operation and Classical Communication (LOCC) theorem [43]. The LOCC states that the two quantum systems cannot be entangled via a classical channel if they were not entangled to begin with, or entanglement cannot be increased by local operations and classical communication. Therefore, by witnessing the entanglement between the two masses, specifically by detecting quantum correlations between the spins that are embedded in the two test masses, we can ascertain whether the gravity is a classical or a quantum entity.

In the QGEM framework, the two free-falling particles thus interact gravitationally in an ostensibly quantum regime—each particle is placed in a superposition and acts as a *quantum source* for the gravitational field. The free-fall of the left (right) particle is determined by the non-classical gravitational field generated by the right (left) particle. We are thus confronted with a free-fall situation that goes beyond classical or quantum EEP in a fixed background gravitational field and therefore requires a novel way of testing the equivalence principle, which we call the entanglement entropy weak equivalence principle (EEWEP).

We define the EEWEP using the relative entanglement entropy generated between the two gravitationally coupled particles (see Equation (Equation 9)). We now first discuss the entanglement entropy in the QGEM scheme.

## 3. Entanglement Entropy

We consider two masses with embedded spins as shown in Figure 1, each of which is placed in a spatial superposition of size Δx. The joint quantum state of the spins $|\Psi (0)\rangle =\frac{1}{2}\left(|↑,↑\rangle +|↓,↓\rangle +|↑,↓\rangle +|↓,↑\rangle \right)$ will evolve to
(1)$$\begin{array}{cc} |\Psi (t)\rangle =\frac{1}{2}e^{i\phi } & \left(|↑,↑\rangle +e^{i\Delta \phi_{\mathrm{ent}}}|↑,↓\rangle +|↓,↓\rangle +e^{i\Delta \phi_{\mathrm{ent}}}|↓,↑\rangle \right), \end{array}$$
where ϕ is a global phase, and $\Delta \phi_{\mathrm{ent}}$ is the entanglement phase. It has been shown that the leading order contribution to the entanglement phase is given by [35]
(2)$$\Delta \phi_{\mathrm{ent}}∼\frac{2Gm_{g}^{2}t\Delta x^{2}}{\hbar d^{3}}+\cdots \;,$$
where *t* is the evolution time, *d* is the distance between two test masses, *G* and *ℏ* are Newton’s and Planck’s constants, respectively, and ⋯ contain higher-order corrections (the operator valued gravitational Hamiltonian, which induces the entanglement phase, can be computed using quantum perturbation theory. At the second post-Newtonian order, the Hamiltonian contains momentum-dependent terms that we assume to be negligible. Similarly, we neglect other higher-order curvature effects $∼O(\Delta x^{4})$ by assuming the superposition size is small compared to the distance between the two particles. Such assumptions are reminiscent of the assumption of a localized point particle in the formulation of the classical WEP, which isolates the leading order effect contributing to the acceleration of point particles. In a complete analogy, we are considering the leading order tidal effect shown in Equation (Equation 2) that dominates the generation of the entanglement). The mass $m_{g}$ that appears in Equation (Equation 2) is to be identified with the gravitational mass (i.e., the mass appearing in the coupling to gravity).

The spatial superposition is created by the electromagnetic interaction, namely via the Stern–Gerlach protocol, which involves inhomogeneous magnetic fields, by displacing the particle according to the spin state. For the purpose of illustration, we assume that the size of the superpositions is given by
(3)$$\Delta x∼\frac{f}{m_{i}}\tau_{a}^{2},$$
where *f* is the force used to prepare/recombine the superpositions in a time $\tau_{a}$. The mass $m_{i}$ appearing in Equation (Equation 3) is to be identified with the inertial mass (i.e., the mass in the free-particle Hamiltonian). For the sake of simplicity, we are assuming a simple mechanism for creating Δx. In reality, we have to specify all the details of how the creation and recombination of the trajectories work out. See for details [44,45]. However, these do not affect the inertial mass, and also all the rest of the model parameters drop out in our definition of $\eta_{s}$, to be defined in Section 4.

Combing Equations (Equation 2) and (Equation 3), we then find
(4)$$\Delta \phi_{\mathrm{ent}}∼\frac{2Gtf^{2}\tau_{a}^{4}}{\hbar d^{3}}{\left(\frac{m_{g}}{m_{i}}\right)}^{2}+\cdots ,$$
where ⋯ again contain higher-order corrections. From Equation (Equation 4), we see that the entanglement phase depends on the fundamental constants (*G* and *ℏ*), on parameters that can be controlled by the experimentalists (*t*, $\tau_{a}$, *d* and *f*) and finally on the ratio $m_{g}/m_{i}$.

Of course, the two masses in systems 1 and 2 need not be the same, allowing us to modify the above expression to
(5)$$\Delta \phi_{\mathrm{ent}}∼\frac{2Gtf^{2}\tau_{a}^{4}}{\hbar d^{3}}\left(\frac{m_{g}^{\left(1\right)}}{m_{i}^{\left(1\right)}}\frac{m_{g}^{\left(2\right)}}{m_{i}^{\left(2\right)}}\right)\frac{{\left(m_{i}^{\left(1\right)}+m_{i}^{\left(2\right)}\right)}^{2}}{m_{i}^{\left(1\right)}m_{i}^{\left(2\right)}}\;,$$
where $m_{g}^{\left(j\right)}$ and $m_{i}^{\left(j\right)}$ denote the gravitational and inertial mass of the *j*-particle, respectively. For the purpose of illustration, we take the inertial masses for both the systems to be the same, but the final expression for the relative entanglement entropy in Equation (Equation 10) remains the same also in the case of unequal masses.

It is instructive to compare the phase in Equation (Equation 4) with the phase obtained in the COW experiment. Specifically, the COW phase is given by $\phi_{\mathrm{COW}}=m_{g}g\Delta xt/\hbar$, where $m_{g}$ is the gravitational mass, *g* is the Earth’s gravitational acceleration, and Δx (*t*) is the superposition size (evolution time) [46,47]. We now use again (Equation 3) to obtain
(6)$$\phi_{\mathrm{COW}}=\frac{GMft\tau_{a}^{2}}{\hbar R^{2}}\left(\frac{m_{g}}{m_{i}}\right),$$
where we have inserted $g=GM/R^{2}$ (*M* is the gravitational mass of the Earth, and *R* is the distance between the experiment and the center of the Earth). The COW phase can thus be used to discern between the gravitational and inertial mass, albeit with a different scaling of the ratio $m_{g}/m_{i}$. More importantly, the COW experiment is conceptually different to the situation depicted in Figure 1. The COW phase arises from the classical background gravitational field generated by the Earth, with the experimental setup bound to its surface. Rather, in the case of Equation (Equation 4), the whole experiment is in free-fall (and hence the COW phase is absent), and the gravitational field is sourced by the particles themselves, each of which is prepared in a superposition state.

Using a similar parametrization as in the classical WEP tests, we quantify the deviation from the expected behavior by writing
(7)$$\frac{m_{g}}{m_{i}}\equiv 1+\xi ,$$
where ξ is a dimensionless parameter that can be extracted experimentally. If we have $m_{g}=m_{i}$, then the entanglement phase will not depend on the ratio between the gravitational and inertial mass, i.e., $\Delta \phi_{\mathrm{ent}}=2Gtf^{2}\tau_{a}^{4}/\left(\hbar d^{3}\right)$.

To quantify how the degree of entanglement changes with the parameter ξ, we combine Equations (Equation 4) and (Equation 7) and compute the entanglement entropy $S_{\xi }\left(t\right)$. Assuming that the entanglement phase $\Delta \phi_{\mathrm{ent}}$ is small, we find a simple expression
(8)$$S_{\xi }\left(t\right)=\frac{f^{4}G^{2}\tau_{a}^{8}t^{2}}{2d^{6}\hbar^{2}}{\left(1+\xi \right)}^{4}.$$

By comparing Equations (Equation 4) and (Equation 8), we see that the entanglement entropy is simply the square of the entanglement phase, i.e., $S_{\xi }\left(t\right)∼\Delta \phi_{\mathrm{ent}}^{2}/8$. It is interesting to note that the entanglement entropy is proportional to the $G^{2}/\hbar^{2}$ contribution.

## 4. EEWEP

We can define the *relative* entanglement entropy, similar to the Eötvös parameter,
(9)$$\eta_{s}\left(t\right)\equiv \left|\frac{S-S_{\xi }\left(t\right)}{S+S_{\xi }\left(t\right)}\right|,$$
where $S\equiv S_{0}\left(t_{\mathrm{ref}}\right)$ corresponds to the entanglement entropy with ξ=0 computed at a reference time $t_{\mathrm{ref}}$. Combining Equations (Equation 8) and (Equation 9), we then find
(10)$$\eta_{s}\left(t\right)=\left|\frac{t_{\mathrm{ref}}^{2}-{\left(1+\xi \right)}^{4}t^{2}}{t_{\mathrm{ref}}^{2}+{\left(1+\xi \right)}^{4}t^{2}}\right|,$$
where all the fundamental constants and the experimental parameters on the right-hand side have cancelled, apart from two times, *t* and $t_{\mathrm{ref}}$, and the parameter ξ.

We are thus led to define the entanglement entropy weak equivalence principle (EEWEP): *the relative entanglement entropy generated between two gravitationally coupled particles is independent of their structure or composition*. The EEWEP is related to the Newtonian formulation of the WEP, because the measure $\eta_{s}$ of the EEWEP will provide a value of ξ, which is a measure for the equivalence of inertial and gravitational mass as was the WEP in the Newtonian domain. The EEWEP however provides a reformulation of the Newtonian WEP principle that does not use statements about trajectories, which are debatable in the quantum domain, but provides a purely quantum reformulation of the WEP. Now, $\eta_{s}$ is the measure of EEWEP, and ξ denotes the violation of EEWEP. To measure precisely the EEWEP violation ξ, one requires only measurements of the entanglement entropy $S_{\xi }$ at the times *t* and $t_{\mathrm{ref}}$—one first obtains the value of $\eta_{s}$ from Equation (Equation 9), and then combining it with Equation (Equation 10) extracts the value of the WEP violation ξ. Figure 2 shows the time-dependent evolution of the relative entanglement entropy for different values of ξ.

In the first place, probing the EEWEP violation up to ξ∼10${}^{-2}$, is well within the experimental possibilities, as it only requires temporal control with accuracy ∼10 ms and measurement of entanglement entropy to $S_{\xi }∼O\left(10^{-2}\right)$. For the purpose of illustration, we have also considered experimental values such as $\xi ∼10^{-15}$ just to have an indicative comparison with the sensitivities achieved in the classical WEP tests [3]. Measuring the entanglement entropy with the accuracy $S_{\xi }∼O\left(10^{-15}\right)$ would require exquisite control of the experiment and a large number of experimental runs, which is currently beyond experimental realities. Nonetheless, recent advancements of keeping track of the frequency ratio measurements up to 18-digit accuracy may be the way to track the time evolution of the relative entanglement entropy [48]. Further, note that the time intervals can be controlled with a great precision given the historical achievement of pico-second ($10^{-12}$ s) pulse rise/fall timings with microwave lasers [49], with femto-second timings also achieved more recently—see [50]—which can, in principle, be used to control an interferometer. Quantum tests of the classical WEP (performed by comparing the acceleration between different atoms/isotopes in the earth’s gravitational field) have reached an experimental accuracy of $\xi ∼10^{-7}$ [51,52,53]. Matching this accuracy would require measurements of entanglement entropy with an accuracy of $S_{\xi }∼O\left(10^{-7}\right)$, which have not been realized experimentally.

In principle, one could be able to test the EEWEP for any massive superpositions, but in practice, one is limited to experiments where the interaction between the two particles is dominated by gravity. To witness gravitational-induced entanglement, we need the gravitational interaction to be dominant over other known Standard Model interactions, such as the electromagnetic-induced interactions. This requires a massive superposition of order $m_{A}∼m_{B}∼O\left(10^{-15}-10^{-14}\right)$ kg, Δx∼10−100μm, and d ∼400μm, as suggested in the original QGEM paper [33] and in [54]. These parameters will generate an appreciable graviton-induced entanglement phase and will dominate over the photon induced Casimir–Polder potential for neutral masses. The mass range of the quantum system is such that it effectively modifies the graviton vacuum by less than one-graviton excitation [55,56], and the emission of any gravitational waves is indeed negligible [57,58].

There are however still many experimental challenges: cooling the nano-crystal [59,60,61], creating superpositions [44,62,63,64,65,66,67,68], tackling decoherence (both from “standard” sources of decoherence such as collisions with air molecules [33,54,69,70,71] as well as from the gravitational coupling to classical or quantum detectors [72,73]), as well as controlling noise sources [74].

## 5. Discussion

To summarize, we have provided the first ever quantum protocol to probe the concept of a free-fall in the framework of perturbative quantum gravity coupled to the non-relativistic quantum matter. We have used the graviton-induced entanglement between the two particles to define the concept of EEWEP—a bonafide quantum test of a free-fall with quantum sources of gravity. To quantify EEWEP violations, we have introduced the parameter ξ, which measures the difference between the gravitational mass and the inertial mass, and pointed out that the violation of EEWEP within $\xi ∼O(10^{-2})$ can perhaps be tested in a near-future experiment.

There are also future avenues to probe the concept of free-fall with quantum sources of gravity. Typically, in any theory where the constants of nature are replaced by dynamical entities, they will tend to violate the equivalence principle [75]. Therefore, the Brans–Dicke theory of gravity [76] and string theory would violate the weak equivalence principle [77,78]. This is due to the fact that Newton’s constant depends on the running dilaton, which means that in the gravitational sector, there will be new dynamical off-shell degrees of freedom. This will also be the case in the context of higher derivative theories of gravity [79,80] and non-local theories of gravity [81,82,83,84,85]. It will be interesting to study in the future the predictions for the EEWEP violations in such theories.

## Figures and Tables

**Figure 1 entropy-25-00448-f001:**
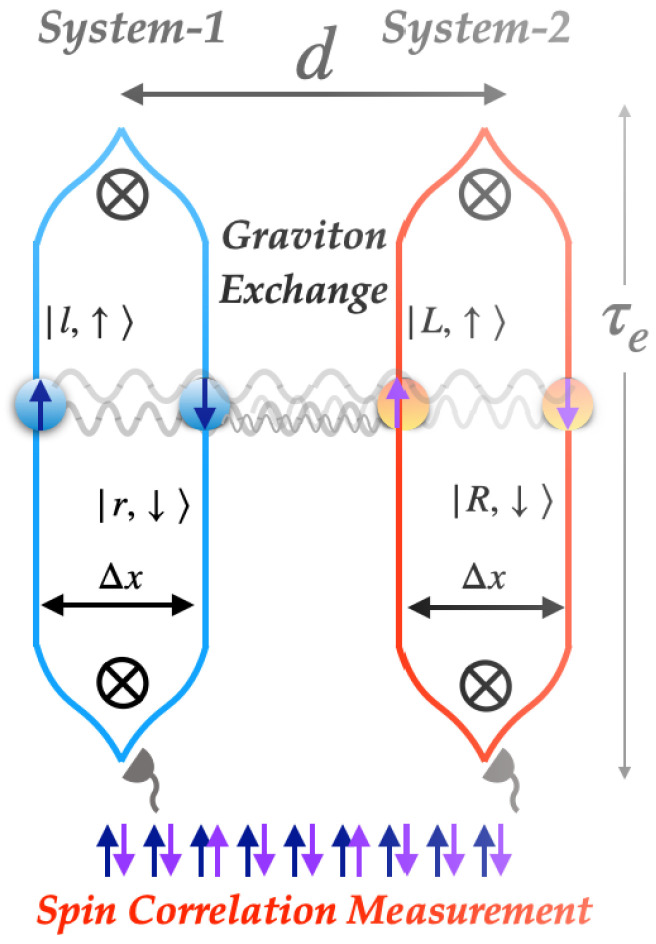
Configuration where the two neutral and massive spatial superpositions are in free-fall with the splitting Δx and separated by a distance *d*. The two spin states (up and down) are embedded in the nano-crystals. The splitting between the two massive spin states is created by an external inhomogeneous magnetic field, similar to the Stern–Gerlach protocol. After the one-loop interference is completed, the spin correlations are computed to witness the entanglement between the two systems induced by the mutual quantum-natured gravitational interaction.

**Figure 2 entropy-25-00448-f002:**
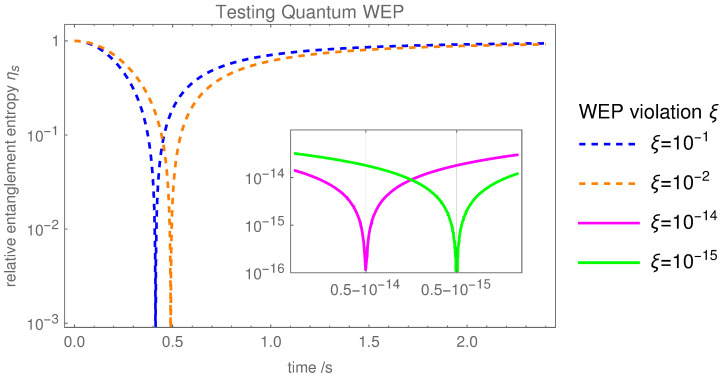
Plot of relative entanglement entropy $\eta_{s}\left(t\right)=\left|\left(S-S_{\xi }\left(t\right)\right)/\left(S+S_{\xi }\left(t\right)\right)\right|$ with respect to the time evolution *t*. $S_{\xi }\left(t\right)$ is the entanglement entropy with the EEWEP violation given by $\xi =m_{g}/m_{i}-1$, and $S=S_{0}\left(t_{\mathrm{ref}}\right)$ is the entanglement entropy without the EEWEP violation computed at the reference time $t_{\mathrm{ref}}=0.5\;s$. Different colours exhibit different values of the EEWEP violation ξ. Testing the EEWEP down to $\xi ∼10^{-2}$ is within experimental possibilities and can be accomplished by measuring the relative entanglement entropy $\eta_{s}$ with accuracy $10^{-2}$. Testing the EEWEP to one part in $10^{2}$ would probe a hitherto unexplored quantum notion of free-fall distinct from any classical test of WEP. The inner embedded plot shows again the relative entropy with respect to time. As an illustration of the scaling of the experimental requirements, we consider the more ambitious value ξ∼10${}^{-15}$. We would require ∼fs resolutions, achievable with atomic clocks, and a scheme for determining the relative entanglement entropy with accuracy $10^{-15}$, the latter being beyond current experimental possibilities.

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
