# Peer review of "Entanglement Witness for the Weak Equivalence Principle"

_entropy, 2023, doi:10.3390/e25030448_

Round 1
Reviewer 1 Report
The authors developed the early very interesting proposal
(itself made by a subset of the authors)
to witness the quantum nature of gravity via nanoparticle
spin correlations, into a test of the weak equivalence principle.
The idea is simple enough, and worth investigating.
However, some issues need to be straightened out.
It should be noted that the weak equivalence principle is not uniquely
defined in quantum mechanics. An interesting account of this issue is
provided in Sakurai's testbook, where the famous COW experiment is
described. Sakurai claims that on one hand the equivalence principle is
violated because "gravity is no longer geometric" up to factors
proportional to A*m/h^2 where A is the characteristic area of the
detector. On the other, he shows that "inertial mass" (which enters
momentum, and hence the De Broglie wavelength) and gravitational mass
are identical, and the COW experiment set limits on this.
While the setup examined by the authors goes "beyond" the COW experiment
as it probes two quatnum masses rather than a quantum test particle in a
gravitational field (the COW experiment does not prove the quantization
of gravity), it is far from clear the limit on the equality of
inertial and gravitational masses is different in these two regimes.
The authors should perhaps compare and contrast on this point.
The authors should also look at experiments such as
https://arxiv.org/abs/1403.1161
A discussion of whether the test proposed here is more stringent than these
other tests would greatly improve this article.
Finally, from the days of Matvei Bronstein, it was realized that
detectors stop being perfect when gravity comes into play.
See https://arxiv.org/abs/2210.08586 for a recent review of this issue.
Does this kind of detector backreaction impact the apparatus as a test of
equality of inertial vs gravitational mass?
Author Response
We thank the referee for the constructive comments which helped us improve our paper.
We have highlighted the key differences between the COW-experiment and the proposed EEWEP experimental scheme in the revised manuscript (see p.4, footnote 3).
The referee pointed out that there exist quantum tests of the violation of the classical WEP. We added Refs. [46-48], and included a brief discussion about these tests, comparing their accuracy to the proposed EEWEP violation test (see L169-L173).
The referee mentioned Bronstein’s discussion about detectors. We have added a small discussion that the presence of physical detectors can lead an intrinsic source of gravitational decoherence. We refer the interested reader to the suggested reference Ref. [65] (see L186-L188).
Reviewer 2 Report
The authors consider the problem of testing the weak equivalence principle in the quantum regime. They do so by developing an entanglement witness based on an experiment implementing the quantum gravity-induced entanglement of masses protocol. The experiment measures the relative phase in the anti-aligned components of two spin-1/2 particles, which is nonzero due to gravitational interactions. The relative phase is a function of the ratio of gravitational-to-inertial mass and, so, can be used to measure deviations from the weak equivalence principle.
The paper is well written, placing their work in the larger context of similar, classical tests and leveraging prior theoretical to design a suitable protocol. Although there is no established quantum theory of gravity, the authors work in a more limited regime in which standard perturbative methods can be applied unabiguously. The methodology is carefully thought out and described, and the authors have made an effort to estimate the degree of violations (or bounds on non-violations) that could be observed.
Testing the weak equivalence principle at these scales could provide valuable new insight into the nature of quantum gravity. However, the experiments will be quite difficult, and those within the realm of feasibility will provide bounds much cruder than those established in the classical regime. Short of an experimentally observed violation, which would be profound, this work is likely to have little impact.
Minor editorial suggestions:
L21: states as follows:
L35: deviations from
L41: wave-functions, which do not
L44: in terms of
L53: pointed out (or pointed to)
L57: would be a major milestone
L74: questions and
L89: character
L94: spins that are embedded
L118: protocol, which involves
L127: , to be defined in Sec. 4.
L142: proportional to the
L157: In the first place
L158: possibilities, as it only
L163: runs, which is
L177: phase and will
Author Response
We appreciate the referee’s positive remarks on our manuscript and for pointing out spelling/grammatical mistakes. We have made the suggested changes in the new version of the manuscript.
Reviewer 3 Report
This is a very nice idea. Definitely worth publishing.
Author Response
We thank referee for their positive assessment on our work.